# CERTIFIABLY ROBUST TRANSFORMERS WITH 1-LIPSCHITZ SELF-ATTENTION

## ABSTRACT

Recent works have shown that neural networks with Lipschitz constraints will lead to high adversarial robustness. In this work, we propose the first One-Lipschitz Self-Attention (OLSA) mechanism for Transformer models. In particular, we first orthogonalize all the linear operations in the self-attention mechanism. We then bound the overall Lipschitz constant by aggregating the Lipschitz of each element in the softmax with weighted sum. Based on the proposed self-attention mechanism, we construct an OLSA Transformer to achieve model *deterministic certified robustness*. We evaluate our model on multiple natural language processing (NLP) tasks and show that it outperforms existing certification on Transformers, especially for models with multiple layers. As an example, for 3-layer Transformers we achieve an $\ell_2$ deterministic certified robustness radius of 1.733 and 0.979 on the word embedding space for the Yelp and SST dataset, while the existing SOTA certification baseline of the same embedding space can only achieve 0.061 and 0.110. In addition, our certification is significantly more efficient than previous works, since we only need the output logits and Lipschitz constant for certification. We also fine-tune our OLSA Transformer as a downstream classifier of a pre-trained BERT model and show that it achieves significantly higher certified robustness on BERT embedding space compared with previous works (e.g. from 0.071 to 0.368 on the QQP dataset).

## 1 INTRODUCTION

Deep neural networks (DNNs) have been widely applied in different domains in recent years, including face recognition (He et al., 2016), machine translation (Bahdanau et al., 2014), and recommendation systems (Zhang et al., 2019b). Specifically, on natural language processing (NLP) tasks, Transformer models (Vaswani et al., 2017) have been proposed and achieved outstanding performance on a variety of tasks. Despite its impressive performance, people have shown that these NLP models suffer from adversarial attacks (Zhang et al., 2020), where an adversary can intentionally inject unnoticeable perturbations on the inputs to fool the model to provide incorrect predictions. Several works have been proposed to improve the empirical robustness of Transformers (Alzantot et al., 2018), but few have studied its certified robustness, i.e. theoretically guarantee that the model will not be attacked under certain conditions (e.g. within some perturbation range). Recently, Shi et al. (2020) proposes to rely on bound-propagation techniques to derive certified robustness for Transformers, which leads to a relatively loose bound and cannot certify on deep models given the looseness induced by propagating from each component in the attention.

In this work, we propose a One-Lipschitz Self-Attention (OLSA) algorithm which provides a robustness certificate for Transformers by bounding the Lipschitz constant of the model. The Lipschitz constant of a model is naturally related to its robustness, as both require that model's output should not change much when the input slightly changes. Previous works (Tsuzuku et al., 2018; Singla and Feizi, 2021) have investigated the 1-Lipschitz property on fully-connected and convolution neural networks, but the study of 1-Lipschitz on Transformer remains unexplored as its complicated non-linear self-attention mechanism are difficult to analyze and constrain. Thus, in this work, we will propose the *first* 1-Lipschitz Transformer network which allows us to achieve tighter deterministic certified robustness against adversarial attacks under different settings (e.g., training from scratch and fine-tuning).

In order to bound the Lipschitz of a self-attention layer, we will first enforce all the linear operations (keys, queries, values) to be orthogonal via re-parametrization techniques (Huang et al., 2020). Next, we will upper bound the input norm by normalizing the word embedding layer. As a result, we will be able to bound the overall Lipschitz by aggregating the change on each component of the softmax weighted sum. Finally, we add scaling factors to ensure 1-Lipschitzness of the OLSA layer. In addition, we also bound the Lipschitz of the pooling layer and aggregate the component to get the final OLSA Transformer classification model.

We evaluate our OLSA Transformer model under both *train-from-scratch* and *fine-tuning* scenarios. In both settings, we show that OLSA achieves significantly higher certified robustness compared with existing bound-propagation-based methods (Shi et al., 2020). The improvement is larger, especially on deeper models. For example, a 3-layer Transformer OLSA Transformer achieves an average certified radius of 1.733 on Yelp, while previous works can only achieve 0.061 under the train-from-scratch setting. When fine-tuning over a BERT pre-trained model, OLSA Transformer achieves a radius of 0.071 on the QQP dataset while previous works can only achieve 0.368. In addition, we show that our certification is 10,000$\times$ faster than previous approaches, since we do not need complicated bound propagation processes and only need one forward pass to perform the certification. Finally, we also evaluate different methods under adversarial attacks and show that OLSA achieves much higher empirical robustness than baselines as well. Meanwhile, we acknowledge a 1% to 2% performance drop on the clean accuracy for OLSA, as we impose the extra 1-Lipschitz constraint which limits the model expressiveness slightly.

**Technical contributions**. We summarize our contributions as follows:

- We propose the first One-Lipschitz Self-Attention mechanism (OLSA) and prove its Lipschitz bound with corresponding analysis.

- We evaluate the proposed OLSA Transformer model on various NLP tasks and observe that it outperforms the state-of-the-art baselines. In particular, the performance gap is larger on deeper models (e.g on 3-layer Transformers on Yelp, we achieve over 25$\times$ average certified radius than previous works).

- The OLSA model requires significantly less time to certify the robustness radius as it only requires a forward pass to calculate the prediction gap.

## 2 RELATED WORK

**Adversarial Robustness for NLP Models** Existing works have shown that NLP models suffer from adversarial attacks (Zhang et al., 2019c; 2020). Adversarial training-based approaches have been proposed to enhance the model robustness during training (Alzantot et al., 2018). In particular, Ren et al. propose to generate adversarial examples with word saliency information. To improve the efficiency of adversarial training, Wang et al. propose a fast gradient projection method. Besides these empirical robustness algorithms, different approaches have been proposed to provide certified robustness on NLP models with smoothing techniques (Ye et al., 2020; Wang et al., 2021a) or bound-propagation techniques (Jia et al., 2019; Shi et al., 2020). However, the smoothing techniques cannot provide deterministic certification, while the bound-propagation techniques are relatively loose and cannot certify for deep models. Recently, (Xu et al., 2020) also proposes a bound-propagation-based technique for NLP models. Their certification is against word substitution attacks and does not directly apply to our scenario.

**Lipschitz-constrained Models and Certified Robustness** The Lipschitz-constrained models have been studied for their smoothness and robustness; however, existing works all focus on constraining the Lipschitz constant for fully-connected and convolutional neural networks. Tsuzuku et al. (2018) first bridges the Lipschitz constant of a network with its robustness and propose a Lipschitz-related loss to improve model robustness. In order to achieve 1-Lipschitzness, multiple works (Cisse et al., 2017; Miyato et al., 2018; Qian and Wegman, 2018) propose to regularize the spectral norm of the linear matrices for fully-connected layers so that the Lipschitz constant is smaller than 1. For convolution neural networks, a simple approach is applied to unroll the convolution into an equivalent linear layer, but this is shown to have a loose Lipschitz bound (Wang et al., 2020). Recent works (Li et al., 2019; Trockman and Kolter, 2021; Singla and Feizi, 2021) have proposed to directly parametrize

a 1-Lipschitz convolutional neural network which achieves a good certified robustness on vision tasks. In addition, existing activation functions such as ReLU are shown not suitable for the 1-Lipschitz models, and better activation functions such as GroupSort (Anil et al., 2019) and Householder (Singla et al., 2021) are proposed. Kim et al. (2021) indeed focuses on the Lipschitz constant of Transformer models. They propose to use $\ell_2$ distance instead of multiplication in the attention mechanism and show that the Lipschitz constant of such variant can be bounded. However, they aim to analyze and provide a (not necessarily tight) upper bound of the Transformer model, but not to propose construction methods for models with small Lipschitz constants.

## 3 BACKGROUND

### 3.1 1-LIPSCHITZ NEURAL NETWORKS AND CERTIFIED ROBUSTNESS

In this work, we focus on the Lipschitz measured under $\ell_2$-norm. We define the Lipschitz constant of a function $f : \mathbb{R}^m \to \mathbb{R}^n$ as :

$$\text{Lip}(f) = \sup \frac{||f(x) - f(x')||_2}{||x - x'||_2} \quad \forall x, x' \in \mathbb{R}^m.$$

We can observe that the Lipschitz property of a neural network is naturally related with its robustness property – both require that the model output should not change much when the input changes by a certain magnitude. Specifically, If we define the prediction margin of $f$ on a certain input $x$ by $\mathcal{M}_f(x) = \max_i f(x)_i - \max_{j \neq \arg\max_i f(x)_i} f(x)_j$ where $f(x)_i$ refers to the indexing operation, then we can guarantee that $f(x)$ will not change its output class within radius $|x - x'| < \mathcal{M}_f(x)/(\sqrt{2}\text{Lip}(f))$. Therefore, people have proposed to enforce 1-Lipschitz for models to achieve robustness. Note that the Lipschitz of a composed function $f \circ g$ satisfies $\text{Lip}(f \circ g) \leq \text{Lip}(f)\text{Lip}(g)$. As a result, we can upper bound the Lipschitz of a model by bounding the Lipschitz of each layer in the neural networks.

### 3.2 ORTHOGONAL LINEAR LAYER IN DNNS

Consider a linear layer $y = Wx$ where $x \in \mathbb{R}^m$, $W \in \mathbb{R}^{n \times m}$, and $y \in \mathbb{R}^n$. People have proposed a stronger constraint to ensure the 1-Lipschitz property: to require that $W$ is orthogonal[1]. The orthogonality not only guarantees that the layer is 1-Lipschitz but also ensures that the gradient norm is preserved during the backward pass, which helps with the training stability (Anil et al., 2019). Several works have proposed re-parametrization techniques to achieve an orthogonal linear layer. For instance, Huang et al. propose to parametrize the orthogonal matrix $W$ with an unconstrained matrix $V \in \mathbb{R}^{n \times m}$ by $W = (VV^\intercal)^{-\frac{1}{2}}V$, where the inverse square root can be calculated by Newton's iteration. In practice, we observe that the following Newton's iteration (Lin and Maji, 2017) achieves a more stable result in calculating the inverse square root, given $Y_0 = VV^\intercal$ and $Z_0 = I$:

$$Y_{k+1} = \frac{1}{2}Y_k(3I - Z_kY_k)$$

$$Z_{k+1} = \frac{1}{2}(3I - Z_kY_k)Z_k$$

$Z$ will converge quadratically to $(VV^\intercal)^{-\frac{1}{2}}$ when $||VV^\intercal - I||_2 < 1$, which in practice is achieved by scaling the parameter $V$ to be small.

Existing works focus on constraining the Lipschitz constant on such linear operation or convolution operation (which is a type of linear operator in a more compact form). However, no one has studied the Lipschitz property of the self-attention mechanism which is a non-linear function. The focus of this paper is to construct the first 1-Lipschitz Transformer model, which we will introduce next.

---

[1]Strictly speaking, we require $W$ to be semi-orthogonal when $n \neq m$, which means either $WW^\intercal = I$ or $W^\intercal W = I$.

## 4 ONE-LIPSCHITZ SELF-ATTENTION (OLSA) TRANSFORMER

As we can see, the property of 1-Lipschitz for neural networks can improve model smoothness and provide certified robustness. However, it is often challenging to enforce the Lipschitz constraint while maintaining a good model capacity. In this section, we will introduce the first 1-Lipschitz Transformer – One-Lipschitz Self-Attention (OLSA) Transformer. We will first introduce the Lipschitz property in a self-attention layer with sequential input and output. Then we introduce how we achieve 1-Lipschitzness for self-attention layers and pooling layers. Finally, we compose these layers and construct the 1-Lipschitz Transformer model.

### 4.1 LIPSCHITZ PROPERTY IN SELF-ATTENTION PIPELINE

First, we describe the setting of the self-attention pipeline as follows. Suppose we have a sequence of input $X \triangleq [x_1, x_2, \ldots, x_N]$ where $x_i \in \mathbb{R}^d$. The self-attention pipeline can be formalized as a function $F : \mathbb{R}^{N \times d} \to \mathbb{R}^{N \times d}$ where the output $Y \triangleq [y_1, y_2, \ldots, y_N] = F(X)$. Under this setting, we define the Lipschitz of the pipeline based on the overall changes in the input sequence:

**Definition 4.1.** Given $X \triangleq [x_1, x_2, \ldots, x_N]$ and $Y \triangleq [y_1, y_2, \ldots, y_N] = F(X)$, we define the Lipschitz of the function as:

$$Lip(F) = \sup_{x_i, x_i'} \frac{\sqrt{\sum_i ||y_i - y_i'||_2^2}}{\sqrt{\sum_i ||x_i - x_i'||_2^2}} = \sup_{X, X'} \frac{||Y - Y'||_F}{||X - X'||_F}$$

where $|| \cdot ||_F$ denotes the Frobenius norm of a matrix.

We can see that such definition of Lipschitz considers the overall potential changes within the input sequence. We aim to bound such Lipschitz constant of the overall model so that we can provide certified robustness for the model against perturbations on the input sequence.

### 4.2 ONE-LIPSCHITZ SELF-ATTENTION (OLSA) LAYER

The standard self-attention mechanism $Y = F(X)$ in a Transformer with input $x_i$'s and output $y_i$'s can be formalized as:

$$s_{ij} = \frac{(W^Q x_i)^\intercal (W^K x_j)}{\sqrt{d}}$$
$$p_{ij} = \text{softmax}([s_{i1}, s_{i2}, \ldots, s_{in}])_j$$
$$y_i = \sum_j p_{ij}(W^V x_j)$$

In order to achieve a tight bound of the Lipschitz constant, we will make two changes to the pipeline. First, we use additive attention instead of dot-product attention in order to provide a tighter bound. As shown in (Vaswani et al., 2017), these two mechanisms do not differ a lot in performance. Second, we add two scaling factors $\alpha_1$ and $\alpha_2$ to control the Lipschitz of the model. The modified self-attention mechanism $Y = F(X)$ is as follows:

$$s_{ij} = \frac{1}{\alpha_1}\left(W^S \sigma(\frac{W^Q x_i + W^K x_j}{2})\right)$$
$$p_{ij} = \text{softmax}([s_{i1}, s_{i2}, \ldots, s_{in}])_j$$
$$y_i = \frac{1}{\alpha_2} \sum_j p_{ij}(W^V x_j)$$

where $\sigma(\cdot)$ denotes some non-linear activation function. After we consider the multi-head mechanism with $H$ headers, the final OLSA layer will be defined as follows:

**Definition 4.2** (OLSA layer). Given $X \in \mathbb{R}^{N \times d}$, the OLSA layer $F : \mathbb{R}^{N \times d} \to \mathbb{R}^{N \times d}$ with $H$ headers is calculated by:

$$s_{ij}^h = \frac{1}{\alpha_1} \left( W_h^S \sigma \left( \frac{W_h^Q x_i + W_h^K x_j}{2} \right) \right)$$

$$p_{ij}^h = \text{softmax}([s_{i1}^h, s_{i2}^h, \ldots, s_{in}^h])_j$$

$$y_i^h = \frac{1}{\alpha_2} \sum_j p_{ij}^h (W_h^V x_j)$$

$$y_i = \text{Concat}([y_i^1, \ldots, y_i^H])$$

where $\{(W_h^S, W_h^Q, W_h^K, W_h^V)\}_{h=1}^H$ are model parameters in which $W_h^S \in \mathbb{R}^{1 \times \frac{d}{H}}$ and $W_h^Q, W_h^K, W_h^V \in \mathbb{R}^{\frac{d}{H} \times d}$.

For this mechanism, we can provide the following Lipschitz bound under the assumption that the linear transformations $W$'s are orthogonal and the input norm is bounded.

**Theorem 4.1** (Lipschitz bound of OLSA layer). *Let $W^Q = Concat([W_1^Q, \ldots, W_H^Q]) \in \mathbb{R}^{d \times d}$ denote the concatenated parameters of $W_h^Q$'s; and $W^K$ and $W^V$ are defined similarly. Assume 1) $W^Q, W^K, W^V$ are orthogonal matrices; 2) $||W_h^S||_2 = 1$ for all $h \in \{1, \ldots, H\}$; 3) $\sigma$ is a 1-Lipschitz activation function; 4) the overall input norm is bounded by $||X||_F \leq c$, then the Lipschitz constant of the function is bounded by:*

$$Lip(F) \leq \frac{1}{\alpha_2} \left( 1 + \frac{c\sqrt{n}}{4\alpha_1} \right).$$

*where $n$ is the length of the input sequence.*

The main idea of the proof is to assume a perturbation $\delta$ on $X$, and gradually bound the perturbation from $s_{ij}^h$'s to $y_i$'s. The main non-linearity comes from the calculation of $y_i^h$ which multiplies the calculated attention score $p_{ij}^h$ with the original input $x_j$. This can be bounded by considering the perturbation on both parts individually, noticing that the norm of both parts is bounded ($p_{ij}^h$ is the output of SoftMax so it is norm-bounded; $x_j$'s norm is bounded in the assumption). The full proof of the theorem is shown in Appendix A. We can observe that the Lipschitz bound of the layer is related to the input sequence length and the input norm bound. We will discuss how we control the input norm for each layer in Section 4.4.

*Remark.* (1) We will train the orthogonal matrices parametrized with $W = (VV^\mathsf{T})^{-\frac{1}{2}} V$ with Newton's iteration (Huang et al., 2020). (2) We would like to get a 1-Lipschitz layer such that the overall Lipschitz of the model is 1, and thus we will set $\alpha_2 = 1 + \frac{c\sqrt{N}}{4\alpha_1}$. (3) We will set $\alpha_1$ to be a trainable parameter. Intuitively, $\alpha_1$ will control the trade-off between self-attention expressiveness and linearity – when $\alpha_1$ is very large, $\alpha_2$ will be close to 1 so that the expressiveness of the final output is preserved, but $s_{ij}$ will all be close to 0, so the attention becomes a simple averaging operation; when $\alpha_1$ is very small, the attention mechanism will work well, but the final output will be divided by a large $\alpha_2$, so the expressiveness of final output is limited.

**Comparison with the Lipschitz bound in (Kim et al., 2021)** In (Kim et al., 2021), the authors also propose a variant of the Transformer L2-MHA and upper bounded its Lipschitz constant with $\alpha = \sqrt{N}(4\phi^{-1}(N-1) + 1) \cdot J(W)^2$, where $J(W)$ is some term related to the spectral norm of $W$ and $\phi^{-1}(N)$ grows slower than $O(\log N)$. Note that they only provide an upper bound of the Lipschitz and do not constrain it to be small. We may also get a 1-Lipschitz model using their bound by orthogonalizing their weight matrices (such that $J(W) = 1$) and re-scaling the output with $1/\alpha$. Theoretically, this bound is $O(\sqrt{N} \log N)$ which is slightly favored than our bound $O(N)^3$. However,

---

[2]In (Kim et al., 2021, Theorem 3.2), the bound has a factor of $\frac{1}{\sqrt{d/H}}$, but this factor comes from the fact that they divide their output $F(X)$ with $\sqrt{d/H}$. For a fair comparison, we do not consider this scaling factor on the output (which is $\alpha_2$ in our case).

[3]Our bound $1 + \frac{c\sqrt{N}}{4}$ is $O(N)$ because we view the embedding norm at each location $||x_i||_2$ to be similar, so the input bound $c = ||X||_F$ is also $O(\sqrt{N})$.

our bound has a significantly smaller constant factors than theirs, as we can observe the factor of 4 in their bound and the factor of $\frac{1}{4}$ in ours. As an example, under a popular case $N = 20$ and input norm is bounded to $||x_i||_2 = 1$, their bound is $\sqrt{N}(4\phi^{-1}(N-1)+1) = 32.28$, while our bound is $1 + \frac{c\sqrt{N}}{4} = 6.0$. Therefore, we can actually provide a tighter Lipschitz guarantee and robustness certification in practice. We include the comparison of using this bound to construct 1-Lipschitz Transformers in Appendix D and observe that OLSA can indeed achieve higher certified robustness than directly adapting (Kim et al., 2021) for certification.

### 4.3 Lipschitz of Pooling Layer

In many tasks, we need a pooling layer after several self-attention mechanisms in order to get a final embedding vector for the entire sequence: $G : \mathbb{R}^{n \times d} \rightarrow \mathbb{R}^d$. The popular approach in standard self-attention layers is to add an extra output token into the sequence and use the embedding of that token for the overall sequence. We intuitively find this approach a "waste of resource" in the 1-Lipschitz case – only one token value is used for pooling while others are dropped, while such information is usually important to achieve a tight bound. In practice, we propose to use the average of all the embeddings of the pooling layer, i.e. $G(X) = \frac{1}{N} \sum_i x_i$. We show that this approach has good Lipschitz properties:

**Theorem 4.2.** *For the pooling function $G(X) = \frac{1}{N} \sum x_i$, we have:*

$$Lip(G) \leq \frac{1}{\sqrt{N}}$$

See Appendix B for the proof. Thus, we can further multiply a factor of $\sqrt{N}$ to get a 1-Lipschitz pooling layer and maximize the output expressiveness. The resulting pooling layer is:

$$G(X) = \frac{1}{\sqrt{N}} \sum_{i=1}^{N} x_i$$

### 4.4 Overall OLSA Transformer

Aggregating different layers we introduced above, we can construct the overall OLSA Transformer, which consists of one word embedding layer, several OLSA layers, one pooling layer and a final linear layer $f : \mathbb{R}^d \rightarrow \mathbb{R}^c$ for prediction, where $c$ is the number of classes. However, there are still some challenges in constructing the OLSA Transformer. The first challenge is how to bound the input norm for each self-attention layer. Note that, the norm of the output layer of self-attention layers will not increase, since it is a weighted average of all the processed input divided by a factor $\alpha_2 > 1$. Therefore, the norm of each layer can be bounded as long as the input to the first layer is bounded. Thus, we will normalize the embedding vector for each token with norm $c$, so that the input to each self-attention layer satisfies $||X||_F \leq \sqrt{N}c$. The second challenge is how to simulate other components with 1-Lipschitz constraint in a standard Transformer. We will remove the LayerNorm and Dropout layers and we use the average $y = 0.5x + 0.5f(x)$ instead of the residual addition connections. We use GroupSort (Anil et al., 2019) as the activation function which is shown to work better than ReLU in 1-Lipschitz networks.

Let $T : \mathbb{R}^{N \times d} \rightarrow \mathbb{R}^c$ denote the OLSA Transformer classification model. We can calculate the certified radius on a given input $X$ with $T(X)$ and $\text{Lip}(T)$. In particular, we can certify that the model prediction will not change within $||X - X'||_F < r$ where:

$$r = \frac{\max_i T(X)_i - \max_{j \neq \arg\max_i T(x)_i} T(X)_j}{\sqrt{2}\text{Lip}(T)}$$

**Comparison with the certification in (Shi et al., 2020)** As the best existing work on robustness certification for Transformers, (Shi et al., 2020) provides certification assuming that only one or two words are perturbed by the adversary. In particular, in their $\ell_2$ certification, they provide certificates that the prediction will not change within $||X - X'||_F < r$, but with the restriction that $X'$ differs with $X$ on only one or two positions of word embedding. Therefore, our work provides a more generalized $\ell_2$ certification. In addition, we explicitly impose the 1-Lipschitz constraint during

training, and therefore the model is optimized to have a better certification radius; by comparison, they will directly certify on a vanilla-trained model, so their certification might not be tight, especially for relatively deep models.

# 5 EVALUATION

In this section, we compare our OLSA Transformer with the state-of-the-art baseline on different datasets under different settings. We observe that our model achieves much higher certified robustness under both train-from-scratch and fine-tuning scenarios. In addition, OLSA is much more efficient during the certification process. We also evaluate our model against the empirical attacks and observe that OLSA also enhances the model empirical robustness.

## 5.1 EXPERIMENTAL SETUP

**Dataset**    We consider three datasets in our evaluation: Yelp (Zhang et al., 2015), SST (Socher et al., 2013) and QQP (Wang et al., 2018). Yelp consists of 560,000/38,000 examples in the training/test set; SST consists of 67,349/872/1,821 examples in the training/development/test set; QQP consists of 363,846/40,430 examples in the training/test set. Each example in Yelp and SST is a sentence labelled with a binary class for its sentiment; each example in QQP consists of two quora questions and labelled with a binary value on whether the two questions are equivalent.

**Implementation Details**    We train and evaluate our 1-Lipschitz OLSA Transformer under both *train-from-scratch* and *fine-tuning* scenarios. In the train-from-scratch scenario, we randomly initialize the model and word embeddings and train it from scratch. In the fine-tuning scenario, we use a pre-trained BERT model (Devlin et al., 2018) and use its output for a downstream OLSA Transformer model. The BERT model is kept unchanged and we only train the downstream model. In the train-from-scratch setting, we use the Yelp and SST datasets which are also adopted in the baseline (Shi et al., 2020). We consider $N$-layer models ($N \leq 3$) and normalize the word embedding and position embedding to a norm of 2, so that the overall input norm is bounded by $||x_i||_2 \leq 4$. For the fine-tuning setting, we evaluate on Yelp, SST and QQP datasets. We normalize the output of BERT to a norm of 2 and use our OLSA Transformer as the downstream model. For both settings, we use the number of attention heads 8 and hidden dimension 256. We train the model with batch size 32 for 50 epochs on SST and 10 epochs on Yelp. For QQP we use Adam optimizer with learning rate $10^{-5}$ and decays by 0.1 at the 40-th epoch. We include certificate regularizer loss (Singla et al., 2021) which adds a term $-\gamma \text{ReLU} \left[ \frac{T(x)_y - \max_{i \neq y} T(x)_i}{\sqrt{2}} \right]$ to maximize the prediction margin with a gradually increasing $\gamma$ with final value $\gamma = 2.0$ for the train-from-scratch case. We use $\gamma = 0.0$ in the fine-tuning scenario as the margin is already large. We will not orthogonalize the final prediction layer; instead, we calculate its Lipschitz and include it in the final certification. For the evaluation time, we compare the time to certify one batch evaluated on an RTX 3090 GPU.

**Baselines**    There are few baselines on the certifiable robustness of Transformer models and (Shi et al., 2020) achieved state-of-the-art certification results. Their model is trained using the standard architecture and training algorithm. We will fix the word embedding to be the same as in our model to perform a fair comparison. To certify the robustness within the region, they propose a bound propagation-based method that tightly bounds the cross-position dependency in the attention mechanism. They evaluate the certification on Yelp and SST datasets on which we will make the comparison and we will also evaluate it on the QQP dataset. Note that since the certification time of their approach is slow, we will only evaluate on a sampled test set with 5% instances for Yelp and QQP and. The comparison with the Lipschitz bound in Kim et al. (2021) is shown in Appendix D, where we adapt their bound to construct the 1-Lipschitz model and observe that our bound provides a better robustness guarantee.

**Evaluation Metrics**    To evaluate the certified robustness, we use the *certified radius* as the metric following the setting in (Shi et al., 2020). Given model $T$ and input $X \in \mathbb{R}^{N \times d}$ on the word embedding space, the certified radius is defined as the maximum perturbation radius within which we

Table 1: *Certified radius* of OLSA and previous state-of-the-art in the train-from-scratch scenario. The certification time is evaluated on a batch of test data.

| Dataset | Depth | Approach | Vanilla Accuracy | Certified radius | Certification Time (sec) | Robust Accuracy under $\ell_2$-**PGD**@$\epsilon = 1.0$ |
|---------|-------|----------|------------------|------------------|--------------------------|-------------------------------------------------------|
| Yelp | 1 | Shi et al. | **89.1%** | 0.476 | 676.8 | 33.5% |
|  |  | OLSA | 87.7% | **1.995** | **0.060** | **61.2%** |
|  | 2 | Shi et al. | **90.3%** | 0.126 | 1164.2 | 26.0% |
|  |  | OLSA | 88.0% | **1.845** | **0.078** | **59.7%** |
|  | 3 | Shi et al. | **90.2%** | 0.061 | 1470.7 | 21.5% |
|  |  | OLSA | 88.1% | **1.733** | **0.105** | **60.1%** |
| SST | 1 | Shi et al. | **81.9%** | 0.923 | 627.8 | 33.1% |
|  |  | OLSA | 80.9% | **1.146** | **0.021** | **54.3%** |
|  | 2 | Shi et al. | **82.5%** | 0.429 | 1056.0 | 33.4% |
|  |  | OLSA | 81.1% | **1.053** | **0.036** | **53.1%** |
|  | 3 | Shi et al. | **82.5%** | 0.110 | 1807.7 | 34.9% |
|  |  | OLSA | 81.2% | **0.979** | **0.049** | **52.3%** |
| QQP | 1 | Shi et al. | **78.8%** | 0.193 | 1527.0 | 6.9% |
|  |  | OLSA | 76.8% | **2.535** | **0.022** | **51.5%** |
|  | 2 | Shi et al. | **79.4%** | 0.083 | 3165.2 | 8.6% |
|  |  | OLSA | 76.6% | **2.383** | **0.037** | **52.6%** |
|  | 3 | Shi et al. | **79.5%** | 0.064 | 4782.1 | 7.1% |
|  |  | OLSA | 77.0% | **2.116** | **0.054** | **47.3%** |

can guarantee that the model prediction will not be changed:

$$\text{Rad}(T, X) = \sup_r r \text{ s.t. } T(X) = T(X') \ \forall ||X - X'||_F \leq r$$

and we calculate the average certified radius over the test dataset.

## 5.2 TRAINING OLSA FROM SCRATCH

We show the results of our certification on the train-from-scratch model in Table 1. We can observe that our OLSA model indeed achieves higher certified robustness compared with Shi et al., especially on deeper layers. We owe it to the reason that the OLSA network is trained to achieve a tight Lipschitz bound and therefore more robust; by comparison, Shi et al. verify on the vanilla model and therefore cannot guarantee a tight bound for each layer. Note that we cannot directly use our certification method to certify their model given that their models do not satisfy 1-Lipschitz so the bound will be very loose. As a cost of improved certified robustness, our models suffer from a 1% to 2% vanilla accuracy drop compared with the vanilla (no regularization) model. We think this is an inevitable performance drop, as shown in previous works (Zhang et al., 2019a) that there exists a trade-off between vanilla accuracy and robustness when comparing vanilla and adversarially robust models. As for the certification time, we can observe that our certification is over 10,000 times faster than previous works. This is because we only need one forward pass to calculate the prediction gap for certification; by comparison, Shi et al. (2020) needs to do a binary search and bound-propagation on each location of the input, which leads to a large number of forward passes.

**Empirical robustness** Besides the certified accuracy, we also perform $\ell_2$-PGD attack (Madry et al., 2017) against the models over the word embedding space to check their empirical robustness as shown in Table 1. We see that OLSA indeed achieves a much higher empirical robustness compared with vanilla models. This confirms that enforcing the 1-Lipschitz constraint indeed helps with the model robustness.

**Ablation studies - Different** $\gamma$ The $\gamma$ in the certificate regularizer loss can control the tradeoff between vanilla accuracy and certified radius — with larger $\gamma$, the model is trained to have a larger

Table 2: Ablation study of a 1-layer OLSA Transformer with different $\gamma$ in the certificate regularizer.

| Dataset | Approach | $\gamma$ | Vanilla Accuracy | Certified radius | Robust Accuracy under $\ell_2$-PGD@$\epsilon = 1.0$ |
|---|---|---|---|---|---|
| | Shi et al. | - | **89.1%** | 0.476 | 33.5% |
| Yelp | OLSA | 0.2 | 90.2% | 1.009 | 53.4% |
| | | 0.5 | 89.9% | 1.274 | 63.3% |
| | | 1.0 | 89.0% | 1.606 | **68.0%** |
| | | 2.0 | 87.7% | 1.995 | 61.2% |
| | | 5.0 | 84.5% | **2.768** | 50.9% |

Table 3: *Certified radius* of 1-layer OLSA and previous state-of-the-arts in the fine-tuning scenario, where we keep the pre-trained BERT model unchanged and only tune the downstream model.

| Dataset | Approach | Vanilla Accuracy | Certified radius | Robust Accuracy under $\ell_2$-PGD@$\epsilon = 1.0$ |
|---|---|---|---|---|
| Yelp (fine-tuning) | Shi et al. | **93.0%** | 0.169 | 59.2% |
| | OLSA | 90.8% | **0.613** | **84.4%** |
| SST (fine-tuning) | Shi et al. | **88.0%** | 0.367 | 52.5% |
| | OLSA | 86.2% | **0.475** | **76.8%** |
| QQP (fine-tuning) | Shi et al. | **83.7%** | 0.071 | 26.2% |
| | OLSA | 81.6% | **0.368** | **66.8%** |

output prediction gap and thus a larger certified radius, at a cost of vanilla accuracy. In Table 2, we show the performance of varying the value of $\gamma$. From the tables, we can indeed observe the trade-off between vanilla accuracy and certified radius. In particular, even with small $\gamma$ we can still achieve a large certified radius, while larger $\gamma$ provides a higher certification result. We set $\gamma = 2.0$ as a reasonable choice for the trade-off. Interestingly, the best empirical robustness is achieved at some intermediate value of $\gamma$. This may be because empirical robustness is not always aligned with certified radius and may be affected by the drop of vanilla accuracy.

## 5.3 FINE-TUNING OLSA OVER PRETRAINED BERT

We show the certified robustness in the fine-tuning scenario in Table 3. The certified robustness is computed over the BERT output embedding space. We mainly evaluate the 1-layer case as it is uncommon to use multi-layer Transformers on top of BERT for downstream tasks (although we expect a larger performance gap for those models). We can see that our OLSA model again achieves a larger certified radius on all three tasks at a small cost of vanilla accuracy. Also, OLSA achieves higher empirical robustness compared with vanilla models. These results show that our OLSA model can be applied in both train-from-scratch and fine-tuning scenarios to enhance model robustness. We provide the training curve in Appendix F. In addition, we evaluate the model robustness on PAWS-QQP (Zhang et al., 2019c), an adversarial dataset of QQP, and also observe that OLSA achieves high empirical robustness, as we show in Appendix C.

## 6 CONCLUSION

In this paper, we propose OLSA, the first 1-Lipschitz self-attention mechanism for sequential input. Based on OLSA layer and 1-Lipschitz pooling, we propose the first 1-Lipschitz Transformer model for NLP classification tasks. We show that OLSA Transformer is able to achieve the state-of-the-art certified robustness on various tasks under both train-from-scratch and fine-tuning scenarios. Our certification is also significantly more efficient. We believe this work will provide a new research direction toward certifiably robust language models.

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

# A    PROOF OF THEOREM 4.1

*Proof.* Consider two inputs $X$ and $X + \Delta X$, where $||X||_F \leq c$, $||X + \Delta X||_F \leq c$ and $||\Delta X||_F \leq \delta$. We will use the symbol $\Delta$ to denote the changes given $X$ and $X + \Delta X$ if it does not lead to confusion. For example, $\Delta Y = F(X + \Delta X) - F(X)$. Our goal is to bound the difference at the output layer $||\Delta Y||_F$. First, we will define the symbols for intermediate results as follows:

$$Q = \text{Concat}([W_1^Q X, \ldots, W_H^Q X]) \in \mathbb{R}^{N \times d}$$
$$K = \text{Concat}([W_1^K X, \ldots, W_H^K X]) \in \mathbb{R}^{N \times d}$$
$$S^h = [s_{ij}^h] \in \mathbb{R}^{N \times N}$$
$$P^h = [p_{ij}^h] \in \mathbb{R}^{N \times N}$$
$$V^h = W_h^V X \in \mathbb{R}^{N \times (d/H)}$$
$$V = \text{Concat}([W_1^V X, \ldots, W_H^V X]) \in \mathbb{R}^{N \times d}$$
$$\tilde{Y}^h = P^h V^h \in \mathbb{R}^{N \times (d/H)}$$

Now, we are going to bound the difference $||\Delta Y||_F$ by considering the perturbations of $\Delta Q$, $\Delta K$, $\Delta S^h$ and $\Delta P^h$ step-by-step.

**Step 1**    $||\Delta Q||_F \leq \delta$, $||\Delta K||_F \leq \delta$ and $||\Delta V||_F \leq \delta$.

We can observe that $Q = W^Q X$, $K = W^K X$ and $V = W^V X$. Since all matrices are orthogonal, the linear operation is 1-Lipschitz. Thus $||\Delta Q||_F \leq ||\Delta X||_F \leq \delta$ and same for $\Delta K$ and $\Delta V$.

**Step 2**    $\sqrt{\sum_h ||\Delta S^h||_2^2} \leq \frac{\sqrt{N}}{\alpha_1} \delta$.

First, notice that the operation $W_h^S \sigma(\cdot)$ is 1-Lipschitz since $||W_h^S||_2 = 1$ and $\sigma$ is 1-Lipschitz. Next, notice that every entry in $Q$ and $K$ will appear $N$ times among the calculation of all the $s_{ij}^h$. Therefore, the overall change of $\sum_{i,j,h}(\Delta s_{ij}^h)^2$ is $N/((2\alpha_1)^2)$ times the overall change of $||\Delta Q||_F$ and $||\Delta K||_F$. So:

$$\sqrt{\sum ||\Delta S^h||_2^2} \leq \sqrt{\frac{N}{(2\alpha_1)^2}(||\Delta Q||_F + ||\Delta K||_F)^2}$$
$$\leq \frac{\sqrt{N}}{\alpha_1} \delta$$

**Step 3**    $\sqrt{\sum_h ||\Delta P^h||_2^2} \leq \frac{\sqrt{N}}{4\alpha_1} \delta$.

This can be proved by noticing that the Lipschitz constant of a SoftMax function is $\frac{1}{4}$.

**Step 4**    $||\Delta \tilde{Y}^h||_F \leq ||V^h||_F + c||P^h||_F$.

Note that the change of $\tilde{Y}^h$ comes from both $V^h$ and $P^h$ and the inequality $||AB||_F \leq ||A||_2||B||_F \leq ||A||_F||B||_F$ holds true. Therefore, we have:

$$\Delta \tilde{Y}^h = P^h \Delta V^h + \Delta P^h (V^h + \Delta V^h)$$
$$\leq ||P^h||_2||\Delta V^h||_F + ||\Delta P^h||_F||V^h + \Delta V^h||_F$$
$$\leq ||\Delta V^h||_F + c||\Delta P^h||_F$$

The last inequality comes from (1) $P^h$ is the matrix after SoftMax, so its spectral norm is no larger than 1; (2) $||V^h + \Delta V^h||_F \leq ||X + \Delta X||_F \leq c$ since $V$ comes from an orthogonal operation of $X$.

**Step 5**    $||\Delta Y||_F \leq \frac{1}{\alpha_2}(1 + \frac{c\sqrt{N}}{4\alpha_1})\delta$.

We notice that $Y = \frac{1}{\alpha_2} \text{Concat}([\tilde{Y}^1, \ldots, \tilde{Y}^H])$. In addition, we have inequality $\sqrt{\sum_h (a_h + b_h)^2} \leq (\sqrt{\sum_h a_h^2} + \sqrt{\sum_h b_h^2})$ from Cauchy's inequality. Therefore, we have:

$$
\begin{aligned}
||\Delta Y||_F &\leq \frac{1}{\alpha_2} \sqrt{\sum_h ||\Delta \tilde{Y}^h||_F^2} \\
&\leq \frac{1}{\alpha_2} \sqrt{\sum_h (||\Delta V^h||_F + c||\Delta P^h||_F)^2} \\
&\leq \frac{1}{\alpha_2} \left( \sqrt{\sum_h ||\Delta V^h||_F^2} + \sqrt{\sum_h (c||\Delta P^h||_F)^2} \right) \\
&\leq \frac{1}{\alpha_2} (1 + \frac{c\sqrt{N}}{4\alpha_1}) \delta.
\end{aligned}
$$

$\square$

## B   PROOF OF THEOREM 4.2

*Proof.* Let $||\Delta X||_F = \delta = \sqrt{\sum_{i=1}^N ||\Delta x_i||_F^2}$ be the perturbation on the input. We have:

$$
\begin{aligned}
||\Delta G(X)||_F &= \frac{1}{N} ||\sum_{i=1}^N \Delta x_i||_F \\
&\leq \frac{1}{N} \sum_{i=1}^N ||\Delta x_i||_F \\
&\leq \frac{1}{N} \sqrt{N \cdot \sum_{i=1}^N ||\Delta x_i||_F^2} \\
&= \frac{1}{\sqrt{N}} \delta
\end{aligned}
$$

where the last inequality is derived from the Cauchy's inequality. $\square$

## C   EVALUATION ON PAWS-QQP

Table 4: The results when we fine-tune on QQP and evaluate on the PAWS-QQP dataset.

| Model | Shi et al. | OLSA ($\gamma = 0$) | OLSA ($\gamma = 0.5$) | OLSA ($\gamma = 2.0$) |
|---|---|---|---|---|
| Vanilla Acc | **83.7%** | 81.6% | 80.3% | 75.4% |
| Robust Acc | 40.9% | 41.9% | 50.8% | **67.7%** |

We show the evaluation results where we fine-tune the model on QQP and evaluate on the adversarial dataset PAWS-QQP Zhang et al. (2019c) in Table 4. We can observe that our OLSA model still achieves a better performance with arbitrary choice of $\gamma$. In particular, larger $\gamma$ can provide a better adversarial robustness, but it may hurt the vanilla accuracy.

## D   COMPARISON WITH (KIM ET AL., 2021)

We compare our bound with the one in (Kim et al., 2021) as in Table 5 for the train-from-scratch scenario and Table 6 for the finetuning scenario. In particular, we will follow their L2-MHA pipeline and (1) orthogonalize all the linear matrices (2) divide the output by their Lipschitz constant bound in

Table 5: Comparison of certification on 1-Lipschitz model using Kim et al. (2021).

| Dataset | Depth | Approach | Vanilla Accuracy | Certified radius |
|---|---|---|---|---|
| Yelp | 1 | Kim et al. | **87.8%** | 1.048 |
| | | OLSA | 87.7% | **1.995** |
| | 2 | Kim et al. | 87.6% | 0.541 |
| | | OLSA | **88.0%** | **1.845** |
| | 3 | Kim et al. | 87.7% | 0.272 |
| | | OLSA | **88.1%** | **1.733** |
| SST | 1 | Kim et al. | 80.8% | 0.628 |
| | | OLSA | **80.9%** | **1.146** |
| | 2 | Kim et al. | 81.1% | 0.324 |
| | | OLSA | **81.1%** | **1.053** |
| | 3 | Kim et al. | 80.7% | 0.174 |
| | | OLSA | **81.2%** | **0.979** |
| QQP | 1 | Kim et al. | 76.2% | 1.522 |
| | | OLSA | **76.8%** | **2.535** |
| | 2 | Kim et al. | 76.0% | 0.771 |
| | | OLSA | **76.6%** | **2.383** |
| | 3 | Kim et al. | 75.6% | 0.413 |
| | | OLSA | **77.0%** | **2.116** |

Table 6: Comparison of certification on 1-Lipschitz pretrained model using Kim et al. (2021).

| Dataset | Approach | Vanilla Accuracy | Certified radius |
|---|---|---|---|
| Yelp (finetuning) | Kim et al. | 88.6% | 0.458 |
| | OLSA | **90.8%** | **0.613** |
| SST (finetuning) | Kim et al. | 85.8% | 0.159 |
| | OLSA | **86.2%** | **0.475** |
| QQP (finetuning) | Kim et al. | 81.1% | 0.251 |
| | OLSA | **81.6%** | **0.368** |

Table 7: *Certified radius* of 1-layer OLSA and previous state-of-the-arts in the fine-tuning scenario, where we use the 11-layer BERT output for OLSA.

| Dataset | Approach | Vanilla Accuracy | Certified radius | Robust Accuracy under $\ell_2$-**PGD**@$\epsilon = 1.0$ |
|---|---|---|---|---|
| Yelp (fine-tuning) | Shi et al. | **93.0%** | 0.169 | 59.2% |
|  | OLSA | 90.9% | **0.525** | **80.3%** |
| SST (fine-tuning) | Shi et al. | **88.0%** | 0.367 | 52.5% |
|  | OLSA | 86.8% | **0.410** | **73.4%** |
| QQP (fine-tuning) | Shi et al. | **83.7%** | 0.071 | 26.2% |
|  | OLSA | 81.8% | **0.329** | **60.4%** |

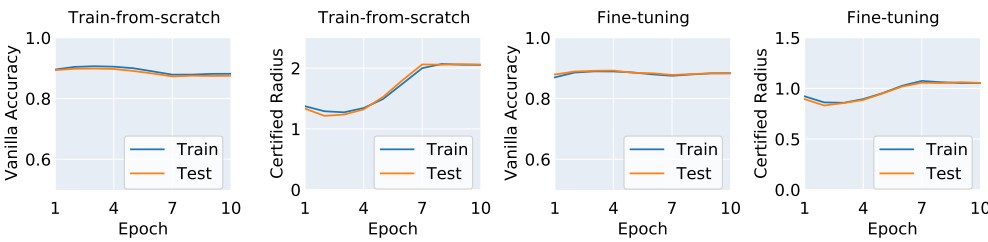

Figure 1: The curve of vanilla accuracy and certified radius of 1-layer OLSA on the Yelp dataset under train-from-scratch and fine-tuning settings.

order to get a 1-Lipschitz self-attention layer. Other settings follow the same way as OLSA. We can observe that our model achieves a larger certified radius as well as a vanilla model accuracy. This is understandable as we discussed that our bound is tighter in practice.

## E PERFORMANCE WITH DIFFERENT LAYERS OF BERT

In the previous evaluation, we use the standard 12-layer BERT architecture followed by a downstream self-attention layer for fine-tuning. In Table 7, we show the results using the output of the 11-th layer in BERT and concatenate it with the downstream layer for finetuning. We view such architecture as a fair analogy of BERT, since it also consists of 12 layers in total. We can observe that there is a slight performance drop compared with the original version, which is understandable as we are only using part of the pretrained BERT. Nevertheless, our model still has a good performance under such circumstance.

## F TRAINING AND CERTIFICATION CURVE

We show the curve of vanilla accuracy and certified radius for the OLSA model on Yelp during training under both scenarios in Figure 1. We have two observations from the curves. First, the model can achieve a good vanilla accuracy at early epochs, while the certified radius is gradually increased during the training process. Note that we have a gradually increasing $\gamma$ which may hurt vanilla accuracy, so the relatively stable accuracy indicates that the model is trained well. Second, we notice that the training and test performance trend is similar as the 1-Lipschitz constraint is a strong regularizer on the model, and thus the model will not overfit on the training set which is desired.

## G ABLATION STUDY

In Table 8, we show the performance of the OLSA network with different features in the tranditional Transformers, including Dropout, ReLU and LayerNorm. In particular, 1) for dropout, we use the PyTorch implementation where we randomly drop out some neurons and scale up other neurons

during training, and perform identity transformation during evaluation; 2) for ReLU, we replace all the GroupSort activation with ReLU activation; 3) for LayerNorm, we only include the bias term and not use the variance term, which is also used in Shi et al. (2020) We can observe that these features have little impact on the final model performance.

Table 8: The results of ablation study on the SST-2 dataset.

| Approach | Vanilla Accuracy | Certified Radius |
|----------|------------------|------------------|
| OLSA     | 80.9%            | 1.146            |
| OLSA     | 80.7%            | 1.142            |
| OLSA     | 80.5%            | 1.174            |
| OLSA     | 80.9%            | 1.144            |

## H  DEEPER MODELS

We provide the results with 4/5/6-layer Transformers on the SST-2 dataset as in Table 9. We can observe that our model can still outperform Shi et al. (2020). In addition, the performance has a slight drop as training deeper models on a single task usually cannot achieve a good performance.

Table 9: The results of deeper models on the SST-2 dataset.

| Depth | Approach | Vanilla Accuracy | Certified Radius |
|-------|----------|------------------|------------------|
| 4     | Shi et al. | 82.7%          | 0.068            |
|       | OLSA     | 81.2%            | 0.919            |
| 5     | Shi et al. | 82.5%          | 0.036            |
|       | OLSA     | 81.3%            | 0.867            |
| 6     | Shi et al. | 82.8%          | 0.021            |
|       | OLSA     | 81.2%            | 0.823            |

## I  EVALUATION ON ADVGLUE

In order to validate the empirical performance of our work, we evaluate our approach over the advGLUE Wang et al. (2021b) dataset, which consists of adversarial examples on SST and QQP generated with different existing attacks. We show the results in Table 10. We can observe that our approach indeed performs better on these adversarially attacked datasets. These results show that Lipschitz is helpful in improving model robustness.

Table 10: The accuracy on the advGLUE dataset.

|            | advSST | advQQP |
|------------|--------|--------|
| Shi et al. | 37.8%  | 57.7%  |
| OLSA       | 41.9%  | 59.0%  |

## J  COMPARISON WITH BONAERT ET AL.

As a recent baseline, Bonaert et al. (2021) improves over Shi et al. (2020) to certify the robustness of a Transformer with bounding propagation techniques. We evaluate their certification approach and provide the comparison of average certified radius in Table 11. Note that the models in this work

are the same as in Shi et al. (2020), so their vanilla accuracy is the same as we show in Table 1. We can observe that our approach still outperforms this work, since their approach is still certifying the vanilla-trained model which intrinsically does not have good robustness.

Table 11: The certified radius comparison with Bonaert et al. on SST and Yelp datasets.

| | 1-layer | 2-layer | 3-layer |
|---|---|---|---|
| Bonaert et al., SST | 0.959 | 0.531 | 0.164 |
| OLSA, SST | 1.146 | 1.053 | 0.979 |
| Bonaert et al., Yelp | 0.875 | 0.317 | 0.204 |
| OLSA, Yelp | 1.995 | 1.845 | 1.733 |

