# OpenReview forum: "Certifiably Robust Transformers with 1-Lipschitz Self-Attention"
_ICLR.cc/2023/Conference — Submitted to ICLR 2023_

### Official Review · Reviewer_XTXc · 2022-10-17

**Confidence:** 4
**Correctness:** 3
**Technical Novelty And Significance:** 3
**Empirical Novelty And Significance:** 3
**Recommendation:** 6

**Clarity, Quality, Novelty And Reproducibility:**

The presentation of this paper is clear and has certain theoretical contributions.

**Strength And Weaknesses:**

Strength:
1. This paper proposes the first One-Lipschitz Self-Attention mechanism (OLSA) and proves the Lipschitz bound.
2. Because the prediction gap is only calculated on a forward pass in the OLSA model, the robustness radius can be certified in much less time.

Weaknesses:
1. Certified defense is usually divided into bound propagation and randomized smoothing (RS). The paper lacks a comparison with the RS method.
2. The method is not evaluated on TectFooler, BERT-Attack and TextBugger.


**Summary Of The Paper:**

This paper introduces Lipschitz constraints into the transformer for higher adversarial robustness. The author proposes the first One-Lipschitz Self-Attention (OLSA) mechanism for Transformer models. The author orthogonalizes all linear operations in the self-attention mechanism and introduces Lipschitz constraints. The proposed method also provides certified guarantees. Experiments demonstrate the effectiveness of this method.

**Summary Of The Review:**

We tend to accept this paper because the method is effective and the presentation is clear.

---

> ### Author Response · Authors · 2022-11-15
> **Response to Reviewer XTXc**
>
> We thank the reviewer for appreciating our paper and providing valuable feedback. We provide our response below.
>
> **Comparison with RS** (Certified defense is usually divided into bound propagation and randomized smoothing (RS). The paper lacks a comparison with the RS method.)
>
> **Response**: We thank the reviewer for pointing to the RS-based methods. As we discussed in Section 2, there are indeed multiple works that use RS-based method for certifying NLP robustness. However, smoothing techniques only provide probabilistic certification while we focus on deterministic certification. Therefore, we only compare with bound-propagation techniques in our evaluation. We will make such discussions clear in our revision.
>
> **Defense performance against existing attacks** (The method is not evaluated on TectFooler, BERT-Attack and TextBugger.)
>
> **Response**: We thank the reviewer for pointing to the empirical attacks against NLP models. We note that our work focuses on the certified robustness, which provides a lower bound of accuracy against any attacks under certain constraints.
> In addition, following the reviewer's suggestion, we conduct additional evaluations against these attacks.  In order to evaluate our model over these attacks, we use the advGLUE [a] dataset, which consists of adversarial examples on SST and QQP generated with different attacks such as TextBugger, TextFooler, BERT-Attack and SememePSO. We show the performance of our model and Shi et al. in the table below. We can observe that our approach indeed performs more robustly against these adversarial attacks. We will include the results in the revision of our paper. Thank you for the helpful suggestions!
>
> [a] Wang, Boxin, et al. "Adversarial GLUE: A Multi-Task Benchmark for Robustness Evaluation of Language Models." Thirty-fifth Conference on Neural Information Processing Systems Datasets and Benchmarks Track (Round 2). 2021.
>
> |   |      advSST     |      advQQP     |
> | --- | --- | --- |
> | Shi et al. |  0.378 | 0.577 |
> | OLSA |    **0.419**   |   **0.590**  |

---

### Official Review · Reviewer_noPi · 2022-10-24

**Confidence:** 3
**Correctness:** 3
**Technical Novelty And Significance:** 3
**Empirical Novelty And Significance:** 3
**Recommendation:** 6

**Clarity, Quality, Novelty And Reproducibility:**

Although limited by several constraints, the proposed method seems sufficiently novel to me. The paper is easy to follow and the experiment results are enough to show that the model can achieve better robustness under these constraints.

**Strength And Weaknesses:**

Strength:
1. The proposed method can achieve a strong certified radius and it is straightforward and easy to implement.
2. The paper is well-written and easy to follow. The comparison with related works is comprehensive.
3. Experiments show the good robustness of the proposed model.

Weakness:
1. The so-called fine-tuning experiment is not a standard fine-tuning setting used by modern works.
2. The OLSA-based transformer model doesn't have layer norm and dropout. I suspect that's why the author is using a transformer with only 3 layers. In order to make this work has a broader impact, it's would be great the find an adequate substitute for these two modules and train a transformer model with more layers.
3. The paper didn't explain how the method could be applied to natural language generation tasks.

**Summary Of The Paper:**

The paper proposed a new multi-head self-attention variant -- OLSA. The proposed method can satisfy the 1-Lipschitz constraint under certain conditions. Thus, the author further proposes a Lipschitz constrained Transformer model. The model achieves a strong certified radius and robust accuracy while maintaining a reasonable vanilla accuracy.

**Summary Of The Review:**

Overall the paper proposed an interesting new variant of transformer that can easily satisfy the 1-Lipschitz constraint. The idea of building a robust transformer model could potentially have a big impact on many downstream tasks. However, the current method still has many limitations, especially since it's not compatible with some important components of SOTA models and its scalability is still unclear.

---

> ### Author Response · Authors · 2022-11-15
> **Response to Reviewer noPi**
>
> We thank the reviewer for providing valuable feedback. We provide our response below to the questions below.
>
> **Fine-tuning setting** (The so-called fine-tuning experiment is not a standard fine-tuning setting used by modern works.)
>
> **Response**: We thank the reviewer for referring to the setting in our fine-tuning experiment. In our fine-tuning setting, we only train the last layer and keep the backbone model unchanged. We agree with the reviewer that this is different with many existing fine-tuning settings where the backbone model is also tuned. We plan to rename it as the pretrained-model setting in the revision of our paper.
>
>
> **Impact of other submodules** (The OLSA-based transformer model doesn't have layer norm and dropout. I suspect that's why the author is using a transformer with only 3 layers. In order to make this work has a broader impact, it's would be great the find an adequate substitute for these two modules and train a transformer model with more layers.)
>
> **Response**: We thank the reviewer for referring to the special features in Transformers and the scalability of our model. As for the special features, we would like to point out that there are two differences in designing 1-Lipschitz networks and standard networks. First, standard architectures focus a lot on regularizing the network so that it does not overfit (e.g. with dropout), while in 1-Lipschitz networks we do not need strong regularization because the 1-Lipschitz constraint is already strong enough. Second, standard architectures are carefully designed so that the gradient will propagate in a proper manner (e.g. LayerNorm), while 1-Lipschitz networks have an intrinsic gradient-norm-preserving property (see [a]). Therefore, some features in standard Transformer may not be useful in the design of OLSA. In the Table A below, we show the performance of OLSA network with those features on the SST-2 dataset. In particular, 1) for dropout, we use the PyTorch implementation where we randomly drop out some neurons and scale up other neurons during training, and perform identity transformation during evaluation; 2) for ReLU, we replace all the GroupSort activation with ReLU activation; 3) for LayerNorm, we only include the bias term and not use the variance term, which is also used in Shi et al. These components will still be 1-Lipschitz. We can observe that these features have little impact on the final model performance.
>
> As for the scalability to larger models, we only considered 3-layer Transformers because we are training the Transformer from scratch on the datasets, and it is rare to train a very deep model from scratch with the small datasets we use. Nevertheless, we provide the results with 4/5/6-layer Transformers on the SST-2 dataset as in Table B below. We can observe that our model can still outperform Shi et al. In addition, the performance has a slight drop as training deeper models on a single task usually cannot achieve a good performance.
>
> [a] Li, Qiyang, et al. "Preventing gradient attenuation in lipschitz constrained convolutional networks." Advances in neural information processing systems 32 (2019).
>
> Table A:
>
> |   |      Vanilla Accuracy    |      Certified Radius    |
> | --- | --- | --- |
> | OLSA |   80.9%  |   1.146   |
> | OLSA (dropout=0.1) |   80.7%   |   1.142   |
> | OLSA (ReLU) |   80.5%   |   1.174   |
> | OLSA (layernorm-novar) |   80.9%   |   1.144   |
>
> Table B:
>
> |   |      Vanilla Accuracy    |      Certified Radius    |
> | --- | --- | --- |
> | Shi el al., 4-layer |   **82.7%**   |   0.068   |
> | OLSA, 4-layer |   81.2%   |   **0.919**   |
> | Shi el al., 5-layer |   **82.5%**   |   0.036   |
> | OLSA, 5-layer |   81.3%   |   **0.867**   |
> | Shi el al., 6-layer |   **82.8%**   |   0.021   |
> | OLSA, 6-layer |   81.2%   |   **0.823**   |
>
> **Application to natural language generation** (The paper didn't explain how the method could be applied to natural language generation tasks.)
>
> **Response**: We thank the reviewer for mentioning the natural language generation tasks. In this work, we mainly study classification tasks over natural language because we are focusing on the “certified robustness” of NLP models, which usually is discussed only under the classification setting. In order to apply our method to generation tasks, we could also replace standard Transformer layers with our OLSA layers, but we would need to define new metrics for the “robustness” of generation tasks so that the application is meaningful. We will include the discussion in the revision of the paper and leave this direction as an interesting future work.

---

### Official Review · Reviewer_QwhB · 2022-10-24

**Confidence:** 4
**Correctness:** 4
**Technical Novelty And Significance:** 3
**Empirical Novelty And Significance:** 4
**Recommendation:** 6

**Clarity, Quality, Novelty And Reproducibility:**

The paper is mostly well-written and easy to follow, except a couple of typos. Some bibliographic entries miss the publication year.
While many of the paper's building blocks already come from the relevant literature (e.g., orthogonality for 1-Lipschitzness), the work is original as it is the first example of 1-Lipschitz transformers (to the best of my knowledge).
The authors provide code in the supplementary material for reproducibility (though I have not run it).

**Strength And Weaknesses:**

OLTA achieves 1-Lipschitzness via a number of modifications to the standard training process. The authors enforce orthogonality in the self-attention layers (in their additive form), remove dropout and LayerNorm, and employ the GroupSort activation function. While each of the ideas was previously presented in the 1-Lipschitzness literature, this is the first application to training 1-Lipschitz transformers.
In spite of the modifications, the authors show that their method achieves larger certified robustness than the relevant previous work from Shi et al. 2020, with a minor cost in terms of standard accuracy. This is definitely a big strength of the work.

Nevertheless, I have the following comments/questions on the experimental section:
- the authors should compare against the bounds from (Bonaert et al. 2021)[https://files.sri.inf.ethz.ch/website/papers/pldi21-transformers.pdf], which were shown to be more effective than those from Shi et al. 2020;
- the speedup against previous certification methods is repeatedly stated. However, bounding algorithms do not necessarily require an ad-hoc training process, which might be more costly. Could the authors provide information on the training runtimes? What is the overhead of running the Newton's iteration?
- it would be nice to present an ablation study of the non-standard components of the training process on standard accuracy. In other words, how does the use of GroupSort, the lack of dropout or LayerNorm, and the use of additive attention?




**Summary Of The Paper:**

The paper proposes OLSA, an algorithm to train transformers for 1-Lipschitzness in the l2 norm, which is directly linked to robustness to l2 perturbations. The authors show that OLSA attains stronger verified robustness than relevant previous work, at a minimal cost in standard accuracy.

**Summary Of The Review:**

OLTA achieves state-of-the-art deterministic certified accuracy on transformers at a minimal cost in terms of standard accuracy. Nevertheless, the paper could benefit from a more recent baseline (see above), reporting and commenting on training runtimes, and an ablation study.

---

> ### Author Response · Authors · 2022-11-15
> **Response to Reviewer QwhB**
>
> We thank the reviewer for the insightful comments and valuable feedback. We provide our response below.
>
> **Comparison with Bonaert et al.** (the authors should compare against the bounds from (Bonaert et al. 2021))
>
> **Response**: We thank the reviewer for pointing to this related work which we have missed in our discussion. This work is indeed an improvement over Shi et al. We evaluate their certification approach and provide the comparison of average certified radius in the table below. Note that the models in Bonaert et al. are the same as in Shi et al., so their vanilla accuracy is the same as we show in the paper (Table 1). We can observe that our approach still outperforms this work, since their approach is still certifying the vanilla-trained model which intrinsically does not have good robustness.
>
> |   |      1-layer    |      2-layer    |      3-layer    |
> | --- | --- | --- | --- |
> | Bonaert et al., SST |  1.037 | 0.658 | 0.350 |
> | OLSA, SST |  **1.146** | **1.053** | **0.979** |
> | Bonaert et al., Yelp |  0.875 | 0.317 | 0.204 |
> | OLSA, Yelp |  **1.995** | **1.845** | **1.733** |
>
> **Training-time efficiency** (the speedup against previous certification methods is repeatedly stated. However, bounding algorithms do not necessarily require an ad-hoc training process, which might be more costly. Could the authors provide information on the training runtimes? What is the overhead of running the Newton's iteration?)
>
> **Response**: We agree with the reviewer that our approach requires a longer training time compared with bounding-based approaches. We evaluate the training time over the SST-2 dataset and observe that our model takes an average of 226.04 seconds for one training epoch, while standard training algorithms takes 34.73 seconds, so there exists a non-negeligible 6 times training overhead. We will add the information in the revision of the paper. Nevertheless, we would still like to argue that test-time efficiency is a more important metric to consider, as people usually train the model once and certify over a large number of instances.
>
> **Non-standard components** (it would be nice to present an ablation study of the non-standard components of the training process on standard accuracy. In other words, how does the use of GroupSort, the lack of dropout or LayerNorm, and the use of additive attention?)
>
> **Response**: We thank the reviewer for referring to the non-standard components in our model. We would like to first point out that there are two differences in designing 1-Lipschitz networks and standard networks. First, standard architectures focus a lot on regularizing the network so that it does not overfit (e.g. with dropout), while in 1-Lipschitz networks we do not need strong regularization because the 1-Lipschitz constraint is already strong enough. Second, standard architectures are carefully designed so that the gradient will propagate in a proper manner (e.g. LayerNorm), while 1-Lipschitz networks have an intrinsic gradient-norm-preserving property (see [a]). Therefore, some features in standard Transformer may not be useful in the design of OLSA.
>
> In the table below, we show the performance of OLSA network with those features on the SST-2 dataset. In particular, 1) for dropout, we use the PyTorch implementation where we randomly drop out some neurons and scale up other neurons during training, and perform identity transformation during evaluation; 2) for ReLU, we replace all the GroupSort activation with ReLU activation; 3) for LayerNorm, we only include the bias term and not use the variance term, which is also used in Shi et al. These components will still be 1-Lipschitz. We can observe that these features have little impact on the final model performance. For additive attention, the original Transformer paper ([b]) has discussed that additive attention performs similarly with the multiplicative one.
>
> [a] Li, Qiyang, et al. "Preventing gradient attenuation in lipschitz constrained convolutional networks." Advances in neural information processing systems 32 (2019).
>
> [b] Vaswani, Ashish, et al. "Attention is all you need." Advances in neural information processing systems 30 (2017).
>
> |   |      Vanilla Accuracy    |      Certified Radius    |
> | --- | --- | --- |
> | OLSA |   80.9%  |   1.146   |
> | OLSA (dropout=0.1) |   80.7%   |   1.142   |
> | OLSA (ReLU) |   80.5%   |   1.174   |
> | OLSA (layernorm-novar) |   80.9%   |   1.144   |

---

> > ### Comment · Reviewer_QwhB · 2022-11-18
> > **Response to authors**
> >
> > Thank you for your response. While this adequately addressed some of my concerns, I would not be comfortable with increasing the score to 8 (7 is not available). However, I increased my confidence score to 4.
> > I would like the authors to adequately stress the training overhead in a paper revision.

---

### Official Review · Reviewer_UH8A · 2022-10-25

**Confidence:** 2
**Correctness:** 3
**Technical Novelty And Significance:** 3
**Empirical Novelty And Significance:** 2
**Recommendation:** 6

**Clarity, Quality, Novelty And Reproducibility:**

The paper is clear about its methods, theorems, and corresponding proofs. The paper is well-written, and the statements about the comparison between OLSA and previous work are helpful to distinguish their contributions. However, certain details in the model training might make this work not easy to reproduce. If the authors can release the source codes and training scripts, it would be most helpful.

**Strength And Weaknesses:**

Strengths:

1. OLSA has advantages over its baseline, as it has a tighter Lipschitz bound (compared to Kim et al. (2021)) and requires only the forward pass (compared to Shi et al. (2020)). With those improvements, it achieves a higher robustness radius and gets faster than Shi et al. (2020), which is verified on two datasets in both training-from-scratch and fine-tuning scenarios.
2. The modification OLSA makes to the Transformer model makes it easier to bound their 1-Lipschitz constant during training, which is well-designed and works well both theoretically and empirically.

Weaknesses:

1. While the changes on Transformers are necessary for the OLSA to achieve robustness, certain features in Transformers must be abandoned. E.g. only additive attention can be used, while most of the pre-trained transformers rely on the dot-product attention mechanism. LayerNorm is removed, which is important when many layers of transformers are stacked. Those changes might be acceptable for 3-layer transformers, but can be dangerous for larger models.
2. Certain steps to bound the Lipschitz constant have dependencies on the sequence length, such as Theorem 4.1 and 4.2, which means the actual Lipschitz bound would differ for different lengths of the input sequences.

Typo: In the last paragraph before the "technical contributions", the paper says OLSA Transformer achieves a radius of 0.071 ... previous work ... 0.368. Should it be 0.368 and 0.071?

The paper seems to use both n and N symbols to represent the sequence length. E.g. in Theorem 4.1 it uses "n" and the remark below 4.1 uses "N". Can you make them consistent? As a reference, N in Shi et al. represents the number of layers.

Questions

1. I have a general question for this line of work: How is it practically useful to model defense? Lipschitz bound guarantees that the model is robust to slight changes in the embedding space, which is not naturally common in attacks. If one or two words are changed/added/deleted, the embeddings will be significantly changed, then how is Lipschitz helpful?
2. Why do you put the comparison with Kim et al. (2021) in the appendix, and not show their efficiency?

**Summary Of The Paper:**

The paper proposes to use Lipschitz constraints to train certifiable robust transformers. With some modifications to the Transformers models, they bound the Lipschitz constant for each layer, which is related to the robustness of the model. Compared to previous work, their proposed model, named one-Lipschitz self-attention mechanism (OLSA) can achieve tighter bound (therefore larger robustness radius) as well as being more efficient, which is empirically verified on both training-from-scratch and fine-tuning scenarios.

**Summary Of The Review:**

The model proposes a new method that guarantees the Lipschitz bound of the Transformer with a smaller robustness radius and efficient certifications. However, certain modifications to the Transformers are necessary under their settings, which could be harmful to its usefulness.

---

> ### Author Response · Authors · 2022-11-15
> **Response to Reviewer UH8A [1/2]**
>
> We thank the reviewer for the insightful comments and suggestions. We provide our responses below.
>
> **Certain features are abandoned** (certain features in Transformers must be abandoned. E.g. only additive attention can be used, while most of the pre-trained transformers rely on the dot-product attention mechanism. LayerNorm is removed, which is important when many layers of transformers are stacked. Those changes might be acceptable for 3-layer transformers, but can be dangerous for larger models.)
>
> **Response**: We thank the reviewer for referring to the special features in Transformers and the scalability of our model. As for the special features, we would like to point out that there are two differences in designing 1-Lipschitz networks and standard networks. First, standard architectures focus a lot on regularizing the network so that it does not overfit (e.g. with dropout), while in 1-Lipschitz networks we do not need strong regularization because the 1-Lipschitz constraint is already strong enough. Second, standard architectures are carefully designed so that the gradient will propagate in a proper manner (e.g. LayerNorm), while 1-Lipschitz networks have an intrinsic gradient-norm-preserving property (see [a]). Therefore, some features in the standard Transformer may not be useful in the design of OLSA. In Table A below, we show the performance of the OLSA network with those features on the SST-2 dataset. In particular, 1) for dropout, we use the PyTorch implementation where we randomly drop out some neurons and scale up other neurons during training, and perform identity transformation during evaluation; 2) for ReLU, we replace all the GroupSort activation with ReLU activation; 3) for LayerNorm, we only include the bias term and not use the variance term, which is also used in Shi et al. These components will still be 1-Lipschitz. We can observe that these features have little impact on the final model performance. For additive attention, the original Transformer paper ([b]) has discussed that additive attention performs similarly with multiplicative one.
>
> As for the scalability to larger models, we only considered 3-layer Transformers because we are training the Transformer from scratch on the datasets, and it is rare to train a very deep model from scratch with the small datasets we use. Nevertheless, we provide the results with 4/5/6-layer Transformers on the SST-2 dataset in Table B below. We can observe that our model can still outperform Shi et al. In addition, the performance has a slight drop as training deeper models on a single task usually cannot achieve a good performance.
>
> [a] Li, Qiyang, et al. "Preventing gradient attenuation in lipschitz constrained convolutional networks." Advances in neural information processing systems 32 (2019).
>
> [b] Vaswani, Ashish, et al. "Attention is all you need." Advances in neural information processing systems 30 (2017).
>
> Table A:
>
> |   |      Vanilla Accuracy    |      Certified Radius    |
> | --- | --- | --- |
> | OLSA |   80.9%  |   1.146   |
> | OLSA (dropout=0.1) |   80.7%   |   1.142   |
> | OLSA (ReLU) |   80.5%   |   1.174   |
> | OLSA (layernorm-novar) |   80.9%   |   1.144   |
>
> Table B:
> |   |      Vanilla Accuracy    |      Certified Radius    |
> | --- | --- | --- |
> | Shi el al., 4-layer |   **82.7%**   |   0.068   |
> | OLSA, 4-layer |   81.2%   |   **0.919**   |
> | Shi el al., 5-layer |   **82.5%**   |   0.036   |
> | OLSA, 5-layer |   81.3%   |   **0.867**   |
> | Shi el al., 6-layer |   **82.8%**   |   0.021   |
> | OLSA, 6-layer |   81.2%   |   **0.823**   |
>
>
> **Lipschitz depends on sequence length** (the actual Lipschitz bound would differ for different lengths of the input sequences)
>
> **Response**: It is true that the Lipschitz bound of standard self-attention layers will depend on the sequence length. However, in our OLSA layer, we divide the output with a factor so that the overall Lipschitz bound is guaranteed to be 1.0 for input sequences of any length.

---

> > ### Author Response · Authors · 2022-11-15
> > **Response to Reviewer UH8A [2/2]**
> >
> > **Practicality of the defense** (How is it practically useful to model defense? Lipschitz bound guarantees that the model is robust to slight changes in the embedding space, which is not naturally common in attacks. If one or two words are changed/added/deleted, the embeddings will be significantly changed, then how is Lipschitz helpful?)
> >
> > **Response**: We thank the reviewer for the question on the practicality of the attack. First, we would like to argue that the Lp norm in embedding space is also considered in many existing attacks. For example, most synonym word substitution attacks will consider the synonym words whose word embedding are close in Lp norm in the embedding space [a,b]. Therefore, existing certification works (e.g. Shi el al. in our paper) will consider the defense of Lp norm over embedding spaces to defend against such attacks.
> >
> > Second, we have shown the empirical performance of our model under actual attacks in Appendix C of our paper. In addition, in order to validate the empirical performance of our work, we evaluate our approach over the advGLUE [c] dataset, which consists of adversarial examples on SST and QQP generated with different existing attacks such as TextBugger, TextFooler, BERT-Attack and SememePSO. We show the results of our model and Shi et al. in the table below. We can observe that our approach indeed performs better on these adversarially attacked datasets. These results show that Lipschitz is helpful in improving model robustness. We will include the results in the revision of our paper.
> >
> > [a] Alzantot, Moustafa, et al. "Generating Natural Language Adversarial Examples." EMNLP. 2018.
> >
> > [b] Ren, Shuhuai, et al. "Generating natural language adversarial examples through probability weighted word saliency." Proceedings of the 57th annual meeting of the association for computational linguistics. 2019.
> >
> > [c] Wang, Boxin, et al. "Adversarial GLUE: A Multi-Task Benchmark for Robustness Evaluation of Language Models." Thirty-fifth Conference on Neural Information Processing Systems Datasets and Benchmarks Track (Round 2). 2021.
> >
> >
> > |   |      advSST     |      advQQP     |
> > | --- | --- | --- |
> > | Shi et al. |  0.378 | 0.577 |
> > | OLSA |    **0.419**   |   **0.590**  |
> >
> > **Comparison with Kim et al.** (Why do you put the comparison with Kim et al. (2021) in the appendix, and not show their efficiency?)
> >
> > **Response**: We thank the reviewer for pointing to the comparison with Kim et al. We put the comparison with them in the appendix because of the space limit and the fact that we have done the theoretical calculation in Section 4.2 (**Comparison with the Lipschitz bound in (Kim et al., 2021)**) and show that our bound provides tighter bound than theirs in standard settings. As for efficiency, both will require the orthogonalization operation and the only difference is that they use l2 distance to calculate attention score while we use additive attention. Therefore, their efficiency is similar and we do not observe differences in the runtime of the methods as there exists randomness in runtime. We will include the discussion in the revision of the paper.
> >
> > **Typos**
> >
> > **Response**: We thank the reviewer for pointing out the typos and non-consistent notations. We will fix them in the revision of the paper.

---

> ### Comment · Reviewer_UH8A · 2022-12-04
> **Response to the rebuttal**
>
> Thanks for taking the time to answer my questions and address my concerns. I agree that the defense on sequence problems is theoretically hard, and the existing method has to sacrifice certain features of the transformers. Some questions challenged by my might not be solved easily.
>
> I'd keep my current score. I'm less confident in my assessment than other reviewers, so I would suggest the AC refer to other comments left by other reviewers.

---

### Author Response · Authors · 2022-11-15
**Summary of Results in the Response**

We thank all the reviewers for their insightful comments and valuable feedback. We provide the response to each reviewer individually. In addition, we have added several experiment results and updated our paper following the reviewer’s suggestions to further improve our work. We mark the newly added content in blue color in the revision of our paper. Below is a summary of our response:

* We provide the results of our model with 4, 5 and 6 layers in the response to Reviewer UH8A and noPi and in Table 9 in Appendix H. We can observe that our model still performs well with the deeper architecture.
* We provide the ablation study of our model with Dropout, LayerNorm and replacing GroupSort with ReLU in the response to Reviewer UH8A, QwhB and noPi and in Table 8 in Appendix G. We show that these components will not have an obvious impact on the model performance.
* We provide the empirical evaluation of our model on the adversarial NLP dataset advGLUE in the response to Reviewer UH8A and XTXc and in Table 10 in Appendix I. We can observe that our approach works well against existing empirical adversarial attacks on NLP datasets.
* We provide the comparison of a new baseline Bonaert et al. in the response to Reviewer QwhB and in Table 11 in Appendix J. We can observe that our approach can still outperform this recent baseline.
* We provide the training time comparison between our model and existing works in the response to Reviewer QwhB.
* We discuss the potential of our model in the application of NLP generative model in the response to Reviewer noPi.
* We clarify the difference between our approach and probabilistic certification approaches in the response to Reviewer XTXc.

If the manuscript is accepted, these contents will be merged with the main text in the revision of the paper.

---

### Decision · Program_Chairs · 2023-01-20

**Decision:**

Reject

**Justification For Why Not Higher Score:**

I doubt this paper would be impactful given that similar approaches have been tried and none of them have really been folded into the main transformer.

robustness in NLP is a tricky topic and i don't think certified robustness is important at all for language. paper also fails to explain why it's important. if certified robustness is so important than chatGPT would already be using it.

**Justification For Why Not Lower Score:**

Already a reject. Can't be lower.

**Metareview: Summary, Strengths And Weaknesses:**

This paper proposes One-Lipschitz Self-Attention (OLSA) mechanism for Transformer models that proposes to orthogonalize all linear operations.

I am recommending rejection for the following reasons:
1. reviewers are generally not excited about the paper based on scores.
2. orthogonalizing transformers have already been tried in https://aclanthology.org/2021.acl-short.48/ although not really for robustness.
3. this paper does not consider the cost both in terms of engineering and efficiency of these approaches.
4. the experiments look toy-ish and its quite hard to really imagine the impact of such an approach. most likely large scale transformer models (or even small scale transformers) will likely not adopt this.